# First look at emergency medical technician wellness in India: Application of the Maslach Burnout Inventory in an unstudied population

**Kathryn W. Koval**[1]*, **Benjamin Lindquist**[2], **Christine Gennosa**[3], **Aditya Mahadevan**[4], **Kian Niknam**[2], **Sanket Patil**[5], **G. V. Ramana Rao**[6], **Matthew C. Strehlow**[2], **Jennifer A. Newberry**[2]

1 Department of Emergency Medicine, Medical University of South Carolina, Charleston, SC, United States of America, 2 Department of Emergency Medicine, Stanford University School of Medicine, Palo Alto, CA, United States of America, 3 College of Medicine, Medical University of South Carolina, Anderson, SC, United States of America, 4 University of San Diego, San Diego, CA, United States of America, 5 National Reference Simulation Centre, SGT University, Budhera, Gurugram Haryana, India, 6 GVK Emergency Management and Research Institute, Devar Yamzal, Secunderabad, Telangana, India

* kwk@musc.edu

**Data Availability Statement:** A copy of the survey instrument, univariate, and multivariate regression analyses are included in the supplementary files.

## Abstract

### Introduction

Professional wellness is critical to developing and maintaining a health care workforce. Previous work has identified burnout as a significant challenge to professional wellness facing emergency medical technicians (EMTs) in many countries worldwide. Our study fills a critical gap by assessing the prevalence of burnout among emergency medical technicians (EMTs) in India.

### Methods

This was a cross-sectional survey of EMTs within the largest prehospital care organization in India. We used the Maslach Burnout Inventory (MBI) to measure wellness. All EMTs presenting for continuing medical education between July-November 2017 from the states of Gujarat, Karnataka, and Telangana were eligible. Trained, independent staff administered anonymous MBI-Medical Personnel Surveys in local languages.

### Results

Of the 327 EMTs eligible, 314 (96%) consented to participate, and 296 (94%) surveys were scorable. The prevalence of burnout was 28.7%. Compared to EMTs in other countries, Indian EMTs had higher levels of personal accomplishment but also higher levels of emotional exhaustion and moderate levels of depersonalization. In multivariate regression, determinants of burnout included younger age, perceived lack of respect from colleagues and administrators, and a sense of physical risk. EMTs who experienced burnout were four times as likely to plan to quit their jobs within one year.

Further access to the data is available by request. The Stanford study group has thoroughly considered the ethical implications of sharing de-identified data publicly. Due to the size of our data set, full disclosure is thought to risk the identity of the participants, jeopardizing their anonymity with the employer and others, and compromising the conditions under which they agreed to participate. For data requests, please contact the ethics committee below and cite IRB#41940. Stanford IRB, 1705 El Camino Real, Palo Alto, CA 94306, (650) 724-7141 irbeducation@stanford.edu

**Funding:** The author(s) received no specific funding for this work.

**Competing interests:** The authors have declared that no competing interests exist.

## Conclusion

This is the first assessment of burnout in EMTs in India and adds to the limited body of literature among low- and middle-income country (LMIC) prehospital providers worldwide. Burnout was strongly associated with an EMT's intention to quit within a year, with potential implications for employee turnover and healthcare workforce shortages. Burnout should be a key focus of further study and possible intervention to achieve internationally recognized targets, including Sustainable Development Goal 3C and WHO's 2030 Milestone for Human Resources.

## Introduction

Human capital is arguably the most valuable resource in a health system. Health providers have been shown to directly impact population health outcomes [1,2]. Consequently, the development community has increasingly recognized the importance of the workforce in achieving the Sustainable Development Goals (SDG). SDG 3C calls for a "substantial increase [in] health financing and recruitment, development, training and *retention* of the health workforce in developing countries." In solidarity, the World Health Organization (WHO) has set this same goal as one of six milestones for its Global Strategy on Human Resources for Health 2030 [3].

The burden of workforce shortages in low and middle-income countries (LMIC) continues to be a crisis, and India is no exception [4]. By focusing on the retention and wellness of healthcare providers, there is an opportunity for health delivery organizations to improve quality of care and reduce the need for and cost of introductory training and onboarding [5,6]. One way to address provider wellness and limit attrition is through burnout prevention. The concept of burnout popularized by psychologist Freudenberger in 1974 was initially described as "becoming exhausted by making excessive demands on energy, strength, or resources" resulting in a physical and behavioral syndrome from the workplace. His initial study involved workers at a demanding free clinic in New York City [7]. Maslach et al standardized the measurement of burnout in the late 1970's refining its definition to a state of exhaustion, cynicism, and diminished professional efficacy that results from long-term involvement in work situations that are emotionally demanding [8]. More recent research has focused on clarifying the relationships between stress, burnout, depression, and post-traumatic stress disorder to better understand their distinctions as well as the personality factors and adaptive mechanisms that are protective [9–11].

There is increasing evidence that burnout is common among a range of healthcare providers, however evidence is limited in the prehospital workforce. A US-based MBI (Maslach Burnout Inventory) study of physicians demonstrated emergency physicians had the highest level of burnout at almost 70% compared to physicians of other specialties [12]. Previous research in Indian physicians of all specialties demonstrates a burnout prevalence of 35%-71% [13–15]. Less than fifty studies have examined burnout in the prehospital realm of emergency medical technicians (EMTs). Only a handful of these include EMTs in LMIC. Studies of EMTs internationally, including varied burnout assessments such as the Copenhagen Burnout Inventory, report work-related burnout rates ranging from 19%-84% [16–21]. High intensity work with little control, overwhelming quantity and pace of work, and administrative burdens appear to be key factors contributing to burnout in this "front lines" population [22]. No published studies have measured burnout in Indian EMTs.

Provider burnout has consequences for the patient, the provider, the employer, and the healthcare system. Burnout has been linked to lower quality patient care, including decreased provider productivity, increased medical errors, safety-compromising behavior, and patient

dissatisfaction [23–26]. Burnout in medical personnel has even been associated with the theft of medications and supplies [27]. The impact of occupational burnout often extends beyond the workplace, with an increased risk for insomnia, depression, and marital and family problems [8,28,29]. In multiple employee populations, burnout has been associated with absenteeism, tardiness, and an intention to leave the job (i.e. turnover) [18,30–33]. Though challenging to quantify, each of these employee behaviors places an additional financial burden on an employer and strains the healthcare system [34,35]. Expenses include costs to advertise, hire and train new employees, and pay overtime for replacement staff, as well as a decreased ability to provide services and thus garner revenue.

Having a better understanding of burnout rates and contributing factors can guide personnel policies and improve retention. This is an exploratory descriptive study with the primary objective of determining the prevalence of EMT burnout in India. Secondarily, identifying determinants of burnout may provide targets for wellness interventions, which may in turn limit workforce turnover. We hypothesized that EMTs in India are at high risk for burnout due to physically demanding conditions and the particular stresses of emergency care.

## Methods

### Study design and participants

This was a cross-sectional survey study to determine the prevalence of burnout among EMTs in India as measured by the Maslach Burnout Inventory. Surveys were distributed to a convenience sample of Indian EMT-Basics (as opposed to EMT-Advanced) working for the largest ambulance service in India, GVK EMRI. EMT-Basics with this employer receive 52 days or 450 hours of theory and skills training from their employer prior to working on the ambulance, more than the national requirements [36]. Participants were approached during regularly scheduled continuing medical education courses conducted by their employer. EMTs presenting for continuing medical education between July-November 2017 from Gujarat, Karnataka, and Telangana were eligible to participate. EMTs typically travel twice annually to attend courses in Hyderabad and were scheduled to attend classes prior to determination of study dates. During study windows, all classes were approached for participation. Since burnout in this population has never been studied, baseline prevalence is unknown. Pilot data collected in 70 EMTs in February 2017 suggested a burnout prevalence of 25%. Using the normal approximation to the binomial, we estimated that enrolling 289 EMTs would allow us to estimate the prevalence of burnout in the population with a confidence level of 95% [37].

This study was conducted in accordance with the Declaration of Helsinki. Stanford University's Institutional Review Board (IRB#41940) and the local ethics review committee in India (the research board of GVK Emergency Management and Research Institute (EMRI)) approved the research protocol. Written informed consent was obtained from each participant in his or her native language. Incentives were not offered for participation. Respondent names were not collected to maintain anonymity. EMT instructors read the informed consent aloud, answered questions about how to complete survey, and were then asked to leave the room during survey administration to ensure EMTs anonymity and privacy. Independent study staff proctored, collected, and analyzed the surveys.

### Survey instrument and outcome assessment

Our survey included the Maslach Burnout Inventory, a validated, gold-standard survey instrument [8,12,38]. The MBI measures three components of burnout (emotional exhaustion (EE), depersonalization (DP), and personal accomplishment (PA)) on a seven-point scale ranging from never (0) to everyday (6). Higher scores on EE and DP indicate higher levels of burnout,

while higher scores on PA indicate lower burnout. As advised by MBI survey creators, if respondents omitted more than one answer per MBI component, their survey was not averaged, and summative MBI scores required all questions in each category to be completed. The MBI-Medical Personnel Survey was translated into local Indian languages: Gujarati, Kannada, and Telugu. Surveys were translated and then reverse-translated by a second native speaker. Translators were independent of study. The two translations were then reconciled and interpreted with study staff and a third native speaker to ensure the essence of questions was reflected. The MBI authors granted permission for survey translation. As per the MBI's criteria, we defined burnout as those who received an EE score $\geq$ 27 or DP score $\geq$ 10 [8]. Personal accomplishment is not included in the "burnout" calculation. After a review of burnout associations in the literature, questions were added to assess possible predictors of burnout. These were a mixture of short-answer and a seven-point Likert scale questions.

## Statistical analysis

We used descriptive statistics to examine the distribution of primary and secondary outcomes, and other independent variables around 95% confidence intervals. The Chi-squared, Fisher's exact, and Wilcoxon rank sum tests were used, as appropriate, to make comparisons between grouped data. Univariate and multivariate logistic regressions were used to examine measures of association between EMT burnout and demographic and environmental factors. These factors included age, gender, caste level, religion, plans to quit, work environment (e.g. urban/rural), workplace relationships, perceived respect for work, state of employment, and concern for physical safety at work. Significance was defined as an alpha of 0.05. Combined variables were created for disadvantaged castes (backwards caste, scheduled tribe and scheduled caste), workplace relationships (emergency department personnel, police, and ambulance drivers), and perceived respect (from administrators, family, and community) and dichotomized where appropriate into agree or disagree. Participants who provided incomplete MBI information were not included in this study. Any other pieces of missing data were treated as missing and were not reflected upon resulting univariate or multivariate analyses. Analyses were run using STATA 15/SE for Windows (StataCorp, LP College Station, TX).

## Results

Of the 327 EMTs approached, 314 consented to participate (96%). 18 surveys contained incomplete MBI information, resulting in a final sample of 296 surveys (Fig 1). Each state was well represented: Telangana (112 surveys, 38%), Karnataka (106 surveys, 36%) and Gujarat (78 surveys, 26%) (Table 1). The majority of participants were male EMTs ($n = 215$, 73%), which is reflective of the workforce as a whole. The median age of our sample population was 30 years (IQR 27–32). Respondents had worked a median of 6 years as an EMT (IQR 4–8) and worked almost evenly between rural versus urban environments (45% and 50%, respectively). The majority of sampled EMTs identified as Hindu ($n = 251$, 85%) and were highly educated with almost 77% achieving a university or post-graduate degree. EMTs who reported that they belonged to socioeconomically disadvantaged castes (e.g. backwards caste, scheduled tribe, and scheduled caste) constituted the majority of respondents at 70% ($n = 206$).

Burnout prevalence was 28.7% (95%CI: 23.6–34.2) (Table 2). Gujarat had the highest levels of burnout with a rate of 45% followed by Telangana at 25% and Karnataka at 21%. EMTs who had a high degree of burnout tended to be younger than EMTs with low to moderate burnout (29 vs. 30 years, $p<0.001$) and were more likely to be female (39% vs. 25%, $p = 0.017$). There was no significant difference in burnout rates based on religion, marital status, working environment, education, or caste.

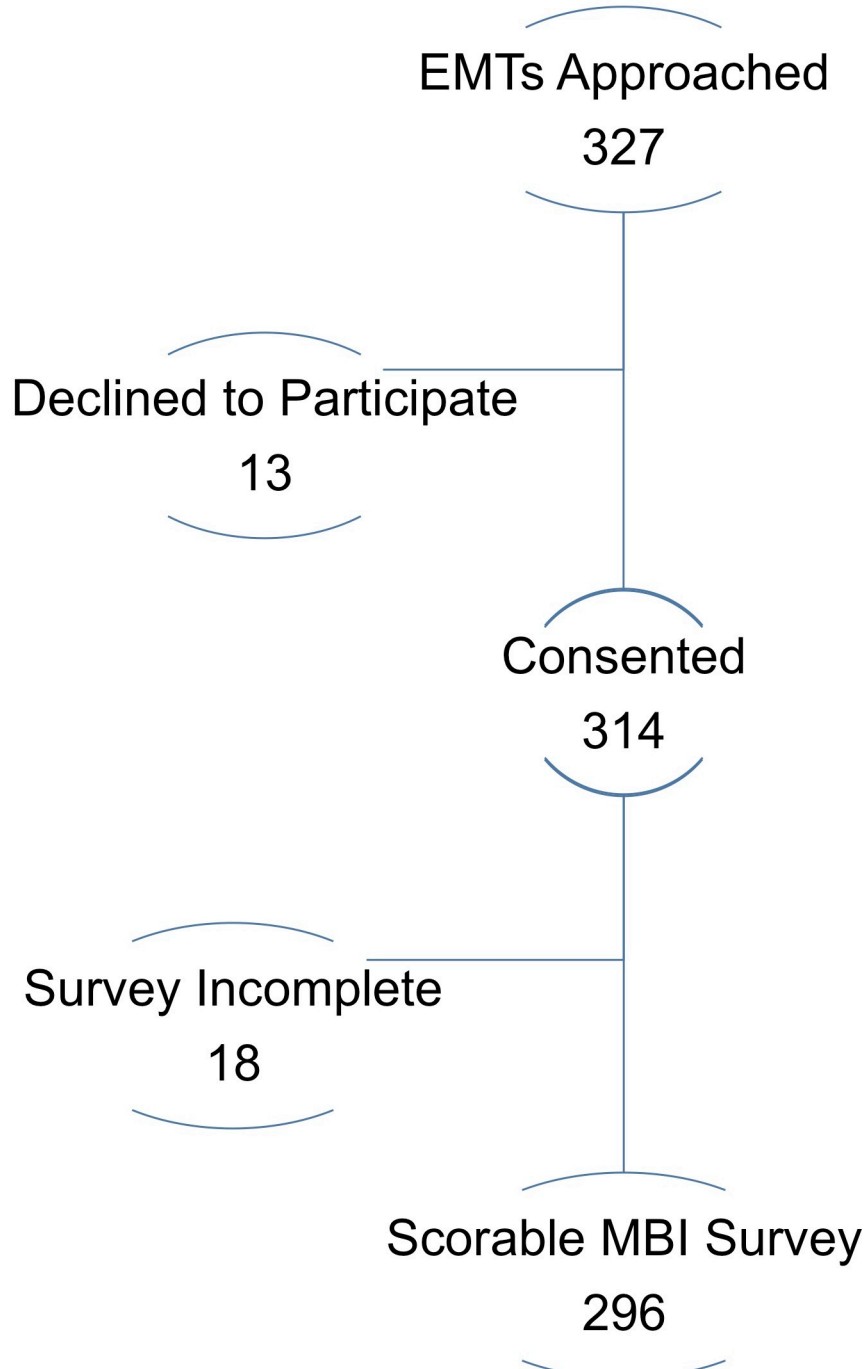

**Fig 1. EMT participation.**

To translate the MBI score to a clinically meaningful endpoint, median average scores for each component are reported: Emotional Exhaustion of 2.0 (IQR: 1.2–2.9), Depersonalization of 1.0 (IQR: 0.4–1.8) and Personal Accomplishment of 5.4 (IQR: 4.8–5.9). These suggest that Indian EMTs experience a sense of personal accomplishment a few times a week, emotional exhaustion once a month or less, and depersonalization a few times of year or less.

**Table 1. Demographics.**

| | |
|---|---|
| Total sample | 296 |
| Median Age [IQR] | 30 [27–32] |
| Median Years Worked [IQR] | 6 [4–8] |
| State *n* (%) | |
| Telangana | 112 (37.8%) |
| Gujarat | 78 (26.4%) |
| Karnataka | 106 (35.8%) |
| Gender *n* (%) | |
| Male | 215 (72.6%) |
| Female | 77 (26.0%) |
| Marital Status *n* (%) | |
| Married | 204 (68.9%) |
| Not Married | 77 (26.0%) |
| Education Level *n* (%) | |
| Below University Degree | 60 (20.2%) |
| University Degree | 170 (57.4%) |
| Post-Grad | 58 (19.6%) |
| Environment *n* (%) | |
| Rural | 134 (45.3%) |
| Urban | 149 (50.3%) |

Percentages may not always add to 100% reflecting occasional missing answers.

 To better compare across populations, median summative MBI scores are also reported (Table 3). MBI scores were able to be totaled for 280 EMTs. The median summative MBI scores were: 18 (IQR: 12–26) for Emotional Exhaustion, 5 (IQR: 2–8) for Depersonalization, and 42 (IQR: 37–47) for Personal Accomplishment. The table is ordered based on emotional exhaustion, the most studied and predictive dimension of burnout [39]. Prehospital personnel have only been surveyed with the MBI in a handful of populations. In this study, Indian EMTs had higher levels of personal accomplishment as compared to Turkish, Spanish, US, and Scottish prehospital personnel. However Indian EMTs experienced more emotional exhaustion than other populations and had moderate levels of depersonalization overall.

## Uni- and multivariate analyses

Secondary outcomes identified demographic and other factors associated with burnout (Table 4). Univariate regression (S1 Table) was performed with clinically or statistically significant factors then included in a multivariate analysis. After controlling for eight independent variables, we observed significant differences in odds of burnout when examining differences in age, perceived lack of respect from administration staff, family, or the community, and feeling physically at risk (Table 4). EMTs who were younger experienced more burnout (OR 0.90, 95% CI = 0.81–0.99, *p* = 0.034) than other EMTs surveyed. Female gender and state appeared to be associated with higher odds of burnout when analyzed individually (OR 1.95, 95% CI 1.12–3.39) (Telengana OR 0.41 (0.22–0.76, *p* = 0.005 / Karnataka OR 0.32, 95% CI 0.17–0.61), but after controlling for other factors, these associations were no longer seen (OR 1.39, 95% CI = 0.47–4.08), (Gujarat OR 0.58, 95% CI 0.28–1.24 / Karnataka OR 0.92, 95% CI 0.28–3.03). In univariate (S1 Table) and multivariate regressions, differences in financial stress, irregular scheduling, experiencing emotionally distressing cases, length of employment, marital status, education level, practice environment, religion, and caste were not predictive of burnout.

**Table 2. Burnout associations with demographics.**

|  | High Degree of Burnout | Low to Moderate Degree of Burnout | p-value |
|---|---|---|---|
| Total sample | 85 (28.7%) | 211 (71.3%) |  |
| Median Age [IQR] | 29 [25–30] | 30 [28–32] | <0.001 |
| Median Years Worked [IQR] | 6 [4–7] | 6 [4–8] | 0.068 |
| State n (%) |  |  | 0.001 |
| Telangana | 28 (25.0%) | 84 (75.0%) |  |
| Karnataka | 22 (20.8%) | 84 (79.2%) |  |
| Gujarat | 35 (44.9%) | 43 (55.1%) |  |
| Gender n (%) |  |  | 0.02 |
| Male | 53 (24.6%) | 162 (75.4%) |  |
| Female | 30 (39.0%) | 47 (61.0%) |  |
| Marital Status n (%) |  |  | 0.57 |
| Married | 54 (26.5%) | 150 (73.5%) |  |
| Not Married | 23 (29.9%) | 54 (70.1%) |  |
| Education Level n (%) |  |  | 0.32 |
| Below University Degree | 15 (25.0%) | 45 (75.0%) |  |
| University Degree | 56 (32.9%) | 114 (67.1%) |  |
| Post-Grad | 13 (22.4%) | 45 (77.6%) |  |
| Environment n (%) |  |  | 0.09 |
| Rural | 31 (23.1%) | 103 (76.9%) |  |
| Urban | 48 (32.2%) | 101 (67.8%) |  |

*Row percentages may not always add to 100 reflecting occasional missing answers

In addition to demographics, EMTs answered questions about their work environment (Table 4). Interpersonal interactions at both work and home influenced the rates burnout. EMTs who did not feel respected by their families, community, and administrators had more than 2 times the odds of burnout (OR 2.30, 95%CI: 1.25–4.26, $p = 0.008$). EMTs who worried for their physical safety while at work were more than 2 times as likely to experience burnout (OR 2.15, 95%CI: 1.11–4.15, p = 0.023). While not significant, EMTs who had poor work relationships with emergency department personnel, police, and ambulance drivers show a strong tendency toward burnout with more than 1.5 times the odds of burnout (OR 1.68, 95%CI: 0.92–3.06, $p = 0.093$).

**Table 3. Comparison of total MBI scores of EMTs across countries˚.**

| Country | Population | n | Emotional Exhaustion˚˚ | Depersonalization˚˚ | Personal Accomplishment˚˚ |
|---|---|---|---|---|---|
| Romania¶ [40] | Paramedic | 258 | 5.7 | 2.3 | 40.1 |
| Spain [17] | Paramedic | 201 | 10.5 | 4.2 | 41.0 |
| USA [41] | EMT, Paramedic, Dispatch | 209 | 13.0 | 6.9 | 39.1 |
| Turkey¶ [42] | Ambulance personnel | 120 | 17.4 | 6.5 | 13.7 |
| Scotland [11] | Ambulance personnel | 110 | 17.2 | 8.4 | 34.5 |
| India | EMT | 280 | 18.0 | 5.0 | 42.0 |
| USA[43] | EMT | 69 | 19.2 | 9.3 | 28.1 |

˚Higher scores on EE and DP indicate higher levels of burnout, while higher scores on PA indicated lower burnout.

˚˚Maximum scores for each category are EE = 54, DP = 30, PA = 48

¶ Scores proportionally adjusted from original study to equally compare MBI components on same scoring scale

**Table 4. Associations with high degree of burnout in multivariate logistic regression model.**

| Characteristic | Odds Ratio | 95% CI | *p*-value |
|---|---|---|---|
| Age | 0.90 | 0.81–0.99 | 0.034 |
| Years worked as EMT | 0.96 | 0.84–1.10 | 0.602 |
| Female gender | 1.39 | 0.47–4.08 | 0.550 |
| Urban work environment | 1.26 | 0.69–2.33 | 0.451 |
| Poor workplace relationships | 1.68 | 0.92–3.06 | 0.093 |
| Perceived lack of respect | 2.30 | 1.25–4.26 | 0.008 |
| Feel physically at risk | 2.15 | 1.11–4.15 | 0.023 |
| State (ref: Telangana) | | | |
| Gujarat | 0.58 | 0.28–1.24 | 0.162 |
| Karnataka | 0.92 | 0.28–3.03 | 0.888 |

Finally, 12% of respondents planned to quit their jobs as EMTs in the next year, and 28% planned to quit within five years. Experiencing burnout was associated with planning to quit within one year; those who were experiencing burnout were 4 times more likely to plan to quit their jobs within a year relative to those who were not burned out (OR: 3.98, 95% CI: 1.49–10.62, *p* = 0.006) (S2 Table). This association did not hold for EMTs who planned to quit in five years (OR 1.63, 95% CI: 0.80–3.35, *p* = 0.18).

## Discussion

This is the first study to examine burnout in Indian EMTs, establishing a prevalence of 28.7%. After controlling for possible confounders, EMTs who were younger than the other EMTs in our sample, EMTs who felt physically unsafe at work, and who perceived a lack of respect for their work were much more likely to experience burnout. EMTs who felt they had poor workplace relationships also had a strong tendency toward burnout. A high degree of burnout was associated with planning to quit work as an EMT within one year.

One other study has assessed provider burnout in India using the MBI in a population of physicians. Using an abbreviated form of the survey with three questions in each category, Langade et al. surveyed 482 medical practitioners in India with a bachelors of medicine or surgery and a minimum of five years of experience [13]. Using this same scoring strategy, the percentage of EMTs with high burnout scores in our study was notably less compared to Indian physicians (Emotional Exhaustion: 13.6% EMT v. 45% physician, Depersonalization 3.6% EMT vs. 66% physician, Lack of Personal Accomplishment: 3.5% EMT v. 87% physician) (S3 Table).

Despite burnout's importance in building a healthcare workforce, only a handful of studies examine burnout in prehospital populations throughout LMICs. A study of 260 prehospital providers in Iran demonstrated 47% of EMTs with high levels of emotional exhaustion and 39% with high levels of depersonalization [20]. A similar study in 140 EMTs in Egypt demonstrated 20% with high emotional exhaustion and 9.3% with high depersonalization [21]. Comparing total MBI scores among prehospital providers in LMICs (Romania and Turkey), Indian EMTs appear to demonstrate a higher sense of personal accomplishment, but higher rates of depersonalization and emotional exhaustion. Romanian paramedics in Popa's study were proposed to have lower burnout measures than other populations due to mandatory periodic psychological exams and the nature of their employment through the army. The particular employer in our study has a strong EMT recognition program, which may be contributing to a higher sense of personal accomplishment among this population. However, prehospital emergency care is still very young in India so recognition and respect for the role of an EMT outside

of the organization (i.e. in the family, community, and healthcare system as a whole) may be lagging.

Social support for EMTs in India had implications for burnout both in the sense of respect from family, community, and administrators as well as relationships with colleagues including police, emergency department staff and ambulance drivers. Significant associations between administrator support and burnout and emotional exhaustion scores have been demonstrated by others [44–47]. Grisby et al surveyed 213 US paramedics and found that poor workplace relationships, in particular, relationships with coworkers and emergency department personnel had the strongest correlation with burnout. A study of EMTs in the Netherlands also demonstrated a significant association between emotional exhaustion and a lack of social support from colleagues [45].

The association between age and burnout is inconsistent throughout the literature. Many studies do not demonstrate a relationship [44], some show an increase in burnout with age [48], while others, including our study, demonstrate less burnout with increasing age [20,41]. In the present study, older EMTs in India may have better strategies to cope with the stresses of work, may be more revered, or older EMTs experiencing burnout may have already left the organization.

Indian EMTs who felt physically at risk were more than twice as likely to experience burnout. This survey question was intentionally broad and could reflect physical threats such as violence, lifting a heavy stretcher, or scene risks like road traffic conditions, difficult to access locations, and industrial accidents. In a separate study, 58% of Indian EMTs reported having experienced some form of physical violence in the past one year [49]. The risk of violence and relation to burnout has been identified in other settings. A study of ambulance staff in Turkey demonstrated a relationship between having experienced a physical attack and higher levels of emotional exhaustion [42]. Physical demands or threats and provider injury at work among ambulance personnel in Norway, Germany, and the United States were also associated with burnout [26,44,48].

One of the most valuable findings of this study was the association between burnout and an EMT's plan to quit working as a prehospital provider. One of the best predictors of an employee quitting their job is their intention to quit [50]. To that end, we asked EMTs how many years he or she planned to remain an EMT. EMTs who are experiencing burnout had almost 4 times the odds of planning to quit within one year relative to those who are not experiencing burnout. This association is present throughout burnout literature thereby linking burnout and employee turnover [18,26,48,51,52]. With its strong association, interventions to decrease EMT burnout and improve EMT wellness may be a modifiable risk factor to decrease employee turnover and address workforce shortages.

Evidenced-based interventions for wellness are limited, with no data on burnout interventions in EMTs. While many purport their strategy to combat burnout [53,54], only a few have been able to demonstrate improved outcomes. Some of these strategies are multifaceted including study of environment specific factors, deduced interventions, and intervention accountability [55,56], as well as single interventions such as art therapy [57], and implementation of meditation strategies and formalized mindfulness-based stress reduction (MBSR) courses [58–60]. These have been shown to reduce burnout and perceived stress while increasing self-compassion and overall satisfaction in a diverse group of healthcare professionals but would need internal validation. Some have found that sleep deprivation contributes to burnout, suggesting that changes to EMT schedules and hours may be a modifiable risk factor to mitigate burnout [61,62]. However in our study, EMT's experiencing burnout did not mind irregular work hours or night shifts any more than EMTs who had low levels of burnout (OR 0.69, 95% CI = 0.41–1.16, p = 0.166). This may be a reflection of the 12-hour shift length that is standard for the organization nationwide.

The factors causing burnout and therefore the solutions to improve EMT wellness are likely different among populations and cultures. A one-size fits-all solution will be impossible to find for EMTs worldwide. Instead we think a successful solution first ensures that baseline human dignity is attended to, and then culturally specific values and needs of local EMTs are identified and addressed to promote resilience in a challenging profession.

## Limitations

Results may underestimate the actual prevalence of burnout. Translation of specific words and concepts of the MBI was particularly challenging, as the figurative concept of burnout is not established in any of the three languages. On initial translation "burnout" was literally understood as a candle losing its flame. A notable strength of this study was the intensive time spent to reconcile translations and best capture the meaning of questions. While the MBI has been used before in Indian healthcare populations [13,15], it has not been officially validated in India. Second, while independent study staff administered surveys, there was suggestion that the EMTs who refused to participate did so out of concern for confidentiality, however, the vast majority of EMTs participated. Our team's extensive experience in India and interviews conducted during our preparatory work suggest there is a cultural tendency in India to give favorable feedback since unfavorable feedback is often perceived as a negative reflection on the individual [63]. The strength of this study is also limited by a convenience sample which may introduce selection bias. While the three states surveyed are well represented, there is notable cultural and administrative diversity between states in India, which may limit the generalizability of these findings to all prehospital personnel in India.

## Conclusions

Emergency medical technicians feel the burden of a demanding job worldwide. In India, more than a quarter of EMTs experience burnout with high levels of emotional exhaustion and depersonalization. Determinants of burnout among Indian EMTs included younger age, concerns for physical safety, and perceived lack of respect for their work. EMTs experiencing burnout were significantly more likely to plan to quit their job within one year. The implications for employee burnout on intention to quit and healthcare workforce shortages should be a key focus of further study. Should the strong association seen in this study and throughout the literature reveal causation, preventing provider burnout would be a prime target to combat workforce shortages and to help achieve Sustainable Development Goals and the WHO's 2030 Milestone for Human Resources.

## Supporting information

**S1 Table. Univariate analyses: Associations with burnout.**
(XLSX)

**S2 Table. Univariate analysis: Burnout as predictive of an EMT's plan to quit.**
(XLSX)

**S3 Table. Comparison of abbreviated MBI between Indian EMTs and physicians as measured by Langade et al [31].**
(XLSX)

**S1 File. English version of survey instrument.**
(PDF)

## Author Contributions

**Conceptualization:** Kathryn W. Koval, Benjamin Lindquist, G. V. Ramana Rao, Matthew C. Strehlow, Jennifer A. Newberry.

**Data curation:** Christine Gennosa, Aditya Mahadevan.

**Formal analysis:** Kathryn W. Koval, Kian Niknam.

**Investigation:** Kathryn W. Koval, Benjamin Lindquist, Christine Gennosa, Aditya Mahadevan, Jennifer A. Newberry.

**Methodology:** Kathryn W. Koval, Benjamin Lindquist, G. V. Ramana Rao, Matthew C. Strehlow, Jennifer A. Newberry.

**Project administration:** Kathryn W. Koval, Benjamin Lindquist, Sanket Patil, Matthew C. Strehlow, Jennifer A. Newberry.

**Resources:** Sanket Patil, G. V. Ramana Rao, Matthew C. Strehlow.

**Supervision:** Kathryn W. Koval, Sanket Patil, G. V. Ramana Rao, Matthew C. Strehlow, Jennifer A. Newberry.

**Validation:** Kathryn W. Koval.

**Visualization:** Kathryn W. Koval, Kian Niknam.

**Writing – original draft:** Kathryn W. Koval.

**Writing – review & editing:** Kathryn W. Koval, Benjamin Lindquist, Christine Gennosa, Aditya Mahadevan, Kian Niknam, G. V. Ramana Rao, Matthew C. Strehlow, Jennifer A. Newberry.

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
