## [Decision Letter · Decision Letter 0]

3 Dec 2019

PONE-D-19-30179

First Look at Emergency Medical Technician Wellness in India: Application of the Maslach Burnout Inventory in an Unstudied Population

PLOS ONE

Dear Dr. Koval,

Thank you for submitting your manuscript to PLOS ONE. After careful consideration, we feel that it has merit but does not fully meet PLOS ONE’s publication criteria as it currently stands. Therefore, we invite you to submit a revised version of the manuscript that addresses the points raised during the review process.

We would appreciate receiving your revised manuscript by Jan 17 2020 11:59PM. To enhance the reproducibility of your results, we recommend that if applicable you deposit your laboratory protocols in protocols.io, where a protocol can be assigned its own identifier (DOI) such that it can be cited independently in the future. For instructions see: http://journals.plos.org/plosone/s/submission-guidelines#loc-laboratory-protocols

We look forward to receiving your revised manuscript.

Kind regards,

Andrew Carl Miller

Academic Editor

PLOS ONE

Journal Requirements:

Additional Editor Comments (if provided):

Thank you for the opportunity to review this important manuscript. We recommend incorporating the reviewer's feedback to improve the manuscript's message and highlight novel areas.

Reviewers' comments:

Reviewer's Responses to Questions

**Comments to the Author**

1. Is the manuscript technically sound, and do the data support the conclusions?

Reviewer #1: Yes

Reviewer #2: Partly

Reviewer #3: Yes

Reviewer #4: Yes

2. Has the statistical analysis been performed appropriately and rigorously? 

Reviewer #1: Yes

Reviewer #2: Yes

Reviewer #3: Yes

Reviewer #4: Yes

3. Have the authors made all data underlying the findings in their manuscript fully available?

Reviewer #1: Yes

Reviewer #2: Yes

Reviewer #3: Yes

Reviewer #4: Yes

4. Is the manuscript presented in an intelligible fashion and written in standard English?

Reviewer #1: Yes

Reviewer #2: Yes

Reviewer #3: Yes

Reviewer #4: Yes

5. Review Comments to the Author

Reviewer #1: Although an interesting, this study is not original nor unique and does not add significantly to the current literature on EMS burnout. As such I do not believe this manuscript should be published in Plos One, but instead should be submitted to a journal more specifically focused on EMS or Indian Health Care.

Page 2 Lines 1-20

Please put abstract into the standard format, with subheadings to include: Introduction, Methods, Results, Conclusion.

Page 2 Line 4

Need to add an introductory/transition sentence.

Need to state that previous studies have identified burnout as an issue for EMS professionals in other countries.

Page 3 Line 45

You also need to include data from Germany which showed that the dimensions emotional exhaustion and depersonalization were positively associated with the safety outcomes injury and safety compromising behavior.

Baier N, Roth K, Felgner S2, Henschke C. BMC Emerg Med. 2018 Aug 20;18(1):24. doi: 10.1186/s12873-018-0177-2.

Burnout and safety outcomes - a cross-sectional nationwide survey of EMS-workers in Germany.

and

also include study from

Nirel N, Goldwag R, Feigenberg Z, Abadi D, Halpern P. Stress, work overload, burnout, and satisfaction among paramedics in Israel. Prehosp Disaster Med. 2008 Nov-Dec;23(6):537-46.

Page 9 Line 133 -135

Please clarify

Other EMS studies on burnout that used the MBI reported data as a Mean with SD. S

Suggest you do the same.

Page 9 Table 3

Suggest you make this into 3 separate tables. Be sure to also include total number in each study for example India N= 327, Romania N=258 etc.

Table 3A Emotional exhaustion

Table 3B Depersonalization

Table 3C Personal accomplishment

Page 9 Table 3

Probable typo from Romanian study.

Not sure where you got your Romanian data since paramedic data from Romania study were not consistent with the Romanian manuscript "The best-recorded values are for paramedics (EE=0.63, DP=0.46, PA=5.01)"

Page 9 Table 3

Typo/Errors regarding data from Turkish study.

The Turkist study has mainly paramedic not EMTs.

Furthermore, the Paramedic sub scores were 8.48 ± 3.91 13.57 ± 5.48 9.56 ± 4.87, not the ones you show.

Page 9 Table 3

Most of the other studies gave results as a mean with standard deviation.

Thus, to better compare your study to the others suggest you include the mean and SD given in the other studies

for Emotional exhaustion, Depersonalization, and Personal accomplishment.

Page 10 Lines 150-153

If possible do not document any results as a p value, but instead use an Odd Ratio with 95% CI.

Currently reads “Younger EMTs experienced more burnout (p=0.034).”

Should read “Younger EMTs experienced more burnout (OR 0.90, 95%CI=0.81 – 0.99)”

Page 12 Line 191

Expand discussion on why Romanian data was different.

“Burnout levels are lower than those found in other studies [30] and the results might be correlated with the fact that paramedical staff is army-enrolled and must pass periodical psychological examinations … not mandatory for any other of the surveyed”

Page 14 Line 231

Please briefly expand the discussion with two sentences on sleep deprivation.

Please add that another potential cause of burnout includes sleep deprivation and 24 hr shifts and whether this is an issue in your study population and include:

Patterson PD, Weaver MD, Frank RC, et al Association between poor sleep, fatigue, and safety outcomes in emergency medical services providers. Prehosp Emerg Care. 2012 Jan-Mar;16(1):86-97. doi: 10.3109/10903127.2011.616261. Epub 2011 Oct 24.

and

Bennett P, Williams Y, Page N, et al. Associations between organizational and incident factors and emotional distress in emergency ambulance personnel. Br J Clin Psychol. 2005;44(Pt 2):215-26.

And your current reference #25

Popa F, Raed A, Purcărea VL, Lală A, Bobirnac G. Occupational Burnout levels in

326 Emergency Medicine – a nationwide study and analysis. J Med Life. 2010;3(3):207–15.

Reviewer #2: Overall this paper does add knowledge to the EMS literature base. Little is known about the rates of burnout in India. This is a very relevant topic - provider wellness is becoming incredibly important to high functioning EMS systems especially in the US where suicide rates are climbing.

I do feel that the paper is placing substantial emphasis on secondary outcomes and making causative conclusions about data found on analyzing the non-primary data points. This can be somewhat misleading to readers as the strength of this evidence is diminished.

16: You don't define LMIC until later in the paper.

64: The US reader would be more familiar with the term EMT-Basic, although national standards have been revised to just "EMT" for this level of care.

65: You mention at the end of this paper that your sample was a convenience sample - can you explain here how these EMTs were selected for study? The way the classes and subjects were chosen may introduce substantial bias into the data.

83-85: It's not clear why these studies were not analyzed. Is that part of the instructions in the MBI or a decision of the authors? If the latter, that's about 5% of your data set being excluded; there may be enough data to alter the statistical significance of some of your results.

85-86 your later discussion mentions the difficulties with translation of the surveys. I think you need to provide much more detail as to how this occurred. Was the translation done by someone blinded to your study? Were they an author on this paper? Was it a translation or an interpretation? Was the product reviewed by others?

102: ambulance pilot implies aeromedical services

115-117: I find this incredibly interesting as degree requirements for EMS providers in the US is a very controversial topic. This is an excellent jumping point for further study based on your data.

118-119: The western reader may not understand these examples given

Table 2:

No and Little evidence of burnout seem to be vastly different concepts in the heading

It's not clear how the P values are tied to the adjacent data. Is it only presented for significant findings?

Table 3:

Why was USA physicians picked as a line for this table? You mention Indian physicians elsewhere in the paper as a reference for burnout but don't include them in the table.

150: Age is a continuous variable. How did you define "younger"?

157:

Are these questions answered part of the standardized survey instrument or was this something additional added as part of the study?

16, 17 170, 177: 223-etc I dont think your data can support this causality you imply. You show association. Theoretically the decision to change professions could cause the results on their survey, no way to tell the difference in a cross-sectional study such as this. Also this doesn't belong in the analysis/methods section.

224: need a reference for this.

Grammar:

Overall needs thorough read over for grammatical errors. Some sentences / sections start with words such as "yet" and "but" which is awkward and needs revision.

Reviewer #3: Thank you for diving into a topic that is rarely discussed, provider well being. It is very important to improve their job satisfaction in order to better serve the public. Please consider this suggestions for your manuscript.

Methods section:

How ere the study staff trained to administer the survey? Did they clarify questions for the participants? If so, this could also be a limitations because different staff might define a question very differently.

Who translated the survey and how did you ensure it was a meaningful translation? There was mention of an "initial" translation in the limitations sections.

Line 16 - You should spell out "LMIC" in the abstract

Linee 26 - Is it "development community" or "developing communities"?

Line 29 - Would add (WHO) after "World Health Organization" in case you need to repeat the term.

Line 33 - Consider changing sentence to: "...continuous to be a crisis and India is no exception."

Line 35 - Consider changing to "...improve quality of care reducing the need for and cost..."

Line 40 - change "and" to a comma: "...providers worldwide, particularly among emergency ..."

Line 42-43 - Add "the" in front of "health care system"

Line 54 - Consider change to "Having a better understanding of burnout rates..."

Line 59 - Consider change to "...demanding conditions and the particular stresses of emergency care."

Line 64 - Consider change to "Indian basic level EMTs..."

Line 67 - Change "As" to "Since"

Line 96-100 - This is kind of a long sentence with multiple uses of "and" - Consider re-writing this sentence or breaking it up.

Line 115-116 - Change sentences to "...almost evenly between rural and urban environments..." "The majority of sampled EMTs identified as ..."

Line 194 Change to "...has a strong EMT recognition program..."

Line 196-197 - Change to "...very young in India so recognition and respect for the role of the EMT outside the organization (i.e. in the family, community and the health care as a whole) may be lagging."

Line - 200-201 - Consider change to "Significant associations between administrator support, emotional exhaustion scores and burnout have also been demonstrated by others [34-36].

Line 204 - Consider change to "... personnel had the strongest correlation with burnout."

Line 204 - Consider change to "A study of EMTs in the Netherlands also demonstrated a significant..."

Line 208 delete "while"

Line 213 - Change "more than two times as likely" to "more than twice as likely"

-------

I hope this recommendations are useful to you. Thank you for allowing me to review this manuscript.

Reviewer #4: GENERAL

Please insert a space between the last word and [reference].

ABSTRACT

Please follow PLOS ONE guidelines for abstract structure and include appropriate sub-headings.

INTRODUCTION

The readers would benefit from an early clarification of definitions and differences between stress and other associated terms. Stress is experienced on a continuum from eustress to burnout syndrome (BOS). In extreme cases, providers may develop signs and symptoms of posttraumatic stress disorder (PTSD).

May consider including some discussion on stress coping mechanisms and their transition for adaptive to maladaptive.

Line 37-39: Please provide reference(s). I would favor clarifying how the definition has evolved over time. The concept of BOS, as first described by Freudenberger in 1974, refers to a protracted course of distress in which one is unable to cope with stressors over an extended period of time, leading to depletion of the body’s defenses and ultimately physical and emotional exhaustion. In 1996, this concept was further refined and BOS was defined as a syndrome of emotional exhaustion, depersonalization, and reduced personal achievement.

Line 40: Please provide some statistics to clarify the scope of the problem.

Lines 43-53: Each of these references refers to physicians and nurses; the statistics, outcomes, and variables that affect these groups may be different than EMTs. Please provide appropriate references for your target study group (EMTs).

METHODS:

Has the MBI been validated in an Indian cohort? If so, please state and provide reference.

Line 64: Clarify what constitutes a “basic” EMT? What is this opposed to?

Please specify that the sample is a convenience sample.

Was any personal information collected or stored?

How/where was the survey announced or advertised? Ideally the survey announcement should be published as an appendix.

Were any incentives offered (eg, monetary, prizes, or non-monetary incentives such as an offer to provide the survey results)?

Indicate whether any methods such as weighting of items or propensity scores have been used to adjust for the non-representative sample; if so, please describe the methods.

RESULTS / DISCUSSION

Line 138-141: This is an interesting observation. Why do you think that is? Please include in discussion.

Line 150: Why do you think younger EMTs were more likely to experience burnout? This is similar to what has been observed in some nursing populations. Please include in discussion.

Line 151: Why do you think women were more likely to experience burnout? This is the opposite of what has been observed in some nursing populations. This has been observed in nurses as well. Please include in discussion.

Line 174-176: Didn’t you state that these findings were not significant after adjusting for controlling factors (line 152-153).

6. PLOS authors have the option to publish the peer review history of their article (what does this mean?). If published, this will include your full peer review and any attached files.

Reviewer #1: Yes: Juan March MD FAEMS FACEP

Reviewer #2: No

Reviewer #3: No

Reviewer #4: Yes: Andrew C. Miller M.D.

---

## [Author Response · Author response to Decision Letter 0]

22 Jan 2020

The response to reviewers has been included in a point-by-point word document uploaded "Response to Reviewers."

---

## [Decision Letter · Decision Letter 1]

19 Feb 2020

First Look at Emergency Medical Technician Wellness in India: Application of the Maslach Burnout Inventory in an Unstudied Population

PONE-D-19-30179R1

Dear Dr. Koval,

We are pleased to inform you that your manuscript has been judged scientifically suitable for publication and will be formally accepted for publication once it complies with all outstanding technical requirements.

With kind regards,

Andrew Carl Miller

Academic Editor

PLOS ONE

Additional Editor Comments (optional):

The authors have satisfactorily answered the reviewers comments.

Reviewers' comments:

Reviewer's Responses to Questions

**Comments to the Author**

1. If the authors have adequately addressed your comments raised in a previous round of review and you feel that this manuscript is now acceptable for publication, you may indicate that here to bypass the “Comments to the Author” section, enter your conflict of interest statement in the “Confidential to Editor” section, and submit your "Accept" recommendation.

Reviewer #3: All comments have been addressed

2. Is the manuscript technically sound, and do the data support the conclusions?

Reviewer #3: Yes

3. Has the statistical analysis been performed appropriately and rigorously? 

Reviewer #3: Yes

4. Have the authors made all data underlying the findings in their manuscript fully available?

Reviewer #3: Yes

5. Is the manuscript presented in an intelligible fashion and written in standard English?

Reviewer #3: Yes

6. Review Comments to the Author

Reviewer #3: I think the authors addressed all of the reviewers' recommendations adequately. Your revisions and additions created a more robust manuscript. I think burnout in the prehospital setting is extremely important as it is in the rest of the healthcare community. Any research in this area is worthy of consideration for publication.

7. PLOS authors have the option to publish the peer review history of their article (what does this mean?). If published, this will include your full peer review and any attached files.

Reviewer #3: No

---

## [Editor Report · Acceptance letter]

26 Feb 2020

PONE-D-19-30179R1 

First Look at Emergency Medical Technician Wellness in India: Application of the Maslach Burnout Inventory in an Unstudied Population 

Dear Dr. Koval:

I am pleased to inform you that your manuscript has been deemed suitable for publication in PLOS ONE. Congratulations! Your manuscript is now with our production department. 

With kind regards,

on behalf of

Dr. Andrew Carl Miller 

Academic Editor

PLOS ONE